**Subject Category:**
Biology (whole organism)

evolution

sea snakes, parietal foramen, head circulation, cutaneous respiration, brain

**Author for correspondence:**
Alessandro Palci
e-mail: alessandro.palci@flinders.edu.au

# Novel vascular plexus in the head of a sea snake (Elapidae, Hydrophiinae) revealed by high-resolution computed tomography and histology

Alessandro Palci[1,2], Roger S. Seymour[3], Cao Van Nguyen[4], Mark N. Hutchinson[1,2], Michael S. Y. Lee[1,2] and Kate L. Sanders[3]

[1]College of Science and Engineering, Flinders University, Adelaide, South Australia 5042, Australia
[2]South Australian Museum, North Terrace, Adelaide, South Australia 5000, Australia
[3]School of Biological Sciences, University of Adelaide, Adelaide, South Australia 5005, Australia
[4]Institute of Oceanography, Vietnam Academy of Science and Technology, Hanoi, Vietnam

AP, 0000-0002-9312-0559; RSS, 0000-0002-3395-0059;
KLS, 0000-0002-9581-268X

Novel phenotypes are often linked to major ecological transitions during evolution. Here, we describe for the first time an unusual network of large blood vessels in the head of the sea snake *Hydrophis cyanocinctus*. MicroCT imaging and histology reveal an intricate modified cephalic vascular network (MCVN) that underlies a broad area of skin between the snout and the roof of the head. It is mostly composed of large veins and sinuses and converges posterodorsally into a large vein (sometimes paired) that penetrates the skull through the parietal bone. Endocranially, this blood vessel leads into the dorsal cerebral sinus, and from there, a pair of large veins depart ventrally to enter the brain. We compare the condition observed in *H. cyanocinctus* with that of other elapids and discuss the possible functions of this unusual vascular network. Sea snakes have low oxygen partial pressure in their arterial blood that facilitates cutaneous respiration, potentially limiting the availability of oxygen to the brain. We conclude that this novel vascular structure draining directly to the brain is a further elaboration of the sea snakes' cutaneous respiratory anatomy, the most likely function of which is to provide the brain with an additional supply of oxygen.

# 1. Introduction

Sea snakes are an extremely successful radiation of elapids that are adapted to a fully marine lifestyle. It has been estimated that true sea snakes diverged from terrestrial elapids about 16 Ma, resulting in 62 known extant species that are ecomorphologically very disparate and collectively span much of the Indo-West Pacific [1–3]. Sea snakes possess several notable adaptations to a marine lifestyle, including sublingual osmoregulatory salt glands, dorsally located external nares that can be sealed by a valve operated by erectile tissue, and the ability to absorb oxygen through their skin [1,4–6]. Interestingly, some sea snakes in the *Hydrophis* group (e.g. *Hydrophis cyanocinctus* and *Hydrophis spiralis*) display a relatively large foramen piercing the parietal bone, in a position that is reminiscent of the pineal foramen of many non-ophidian squamates (lizards) [7]. However, unlike the condition in lizards, there is no indication of the presence of such a foramen on the surface of the skin (figure 1). Such a foramen has been previously figured in the literature on sea snakes [8,9], but its peculiarity was never discussed. A parietal foramen is assumed to be absent in all snakes [7] and has been absent for at least the last 100 Ma, which is the age of the oldest well-known fossil snakes [10]. Therefore, we decided to investigate the possible function of this pineal-like foramen by examining what kind of soft anatomy is associated with it (i.e. nerves, blood vessels, or both). We surprisingly discovered a network of large blood vessels that has never been described before in any snake and that likely represents a novel respiratory adaptation in sea snakes.

# 2. Material and methods

Two live specimens of *H. cyanocinctus* were acquired from commercial fish traders in Vung Tau, Vietnam. The specimens were euthanized and fixed in Vietnam following the procedure outlined in electronic supplementary material, data S1. One of the specimens was microCT scanned at Adelaide Microscopy, University of Adelaide (Adelaide, South Australia), using a Skyscan 1276 (Bruker) microCT scanner, for a preliminary assessment of the presence of the parietal foramen. After identification of the foramen using three-dimensional reconstruction of the skull via Avizo v. 9.0 (Thermo Scientific), the specimen was then microCT scanned again after it was stained in Lugol's iodine solution ($I_2KI$, 7.5% concentration) in order to increase the radio-opacity and contrast of the soft tissues (diceCT; [11]) (electronic supplementary material, data S1). Observation of the diceCT scan data revealed the presence of a complex network of blood vessels that were then segmented in Avizo.

Because diceCT does not distinguish clearly between types of blood vessels (i.e. arteries versus veins) or among some other types of soft tissues, a series of stained histology sections were also obtained from the second specimen. The stains used were: Elastic van Gieson (EVG); a Bielschowsky's modified method (Biel); Haematoxylin and Eosin (H&E) and Alcian Blue/Periodic Acid Schiff (AB/PAS). Full histology protocols are provided in electronic supplementary material, data S1. High-resolution images of the histology sections were taken with a NanoZoomer 2.0HT digital slide scanner (Hamamatsu Photonics). Measurements of skin thickness were taken from scanned histology slides in NDP view v. 2 (Hamamatsu Photonics), while surface area and volumetric measurements were taken from three-dimensional rendered meshes in Avizo.

For comparative purposes, we also analysed diceCT data from a specimen of the terrestrial elapid *Oxyuranus scutellatus* and from two additional sea snakes, *Hydrophis stokesii* and *Aipysurus laevis*. Stained histology sections were obtained from a second specimen of *O. scutellatus* using the same protocols reported above. All specimens in this study are deposited in the collections of the South Australian Museum, Adelaide (SAMA).

# 3. Results and discussion

The diceCT data from *H. cyanocinctus* revealed an intricate network of large blood vessels (modified cephalic vascular network; MCVN) below the skin of the snout and forehead (i.e. above the frontal and parietal bones) (figure 1). While subcutaneous blood vessels are expected to be present in all snakes, the terrestrial *O. scutellatus* and the two sea snakes, *H. stokesii* and *A. laevis*, were found to possess mostly only fine blood vessels under their skin (figure 1 and electronic supplementary material, figure S1 in data S2). The epidermal microstructure is similar to that reported in the congeneric sea snake *Hydrophis platurus* (i.e. mesos layer absent or very thin; [12]) (electronic

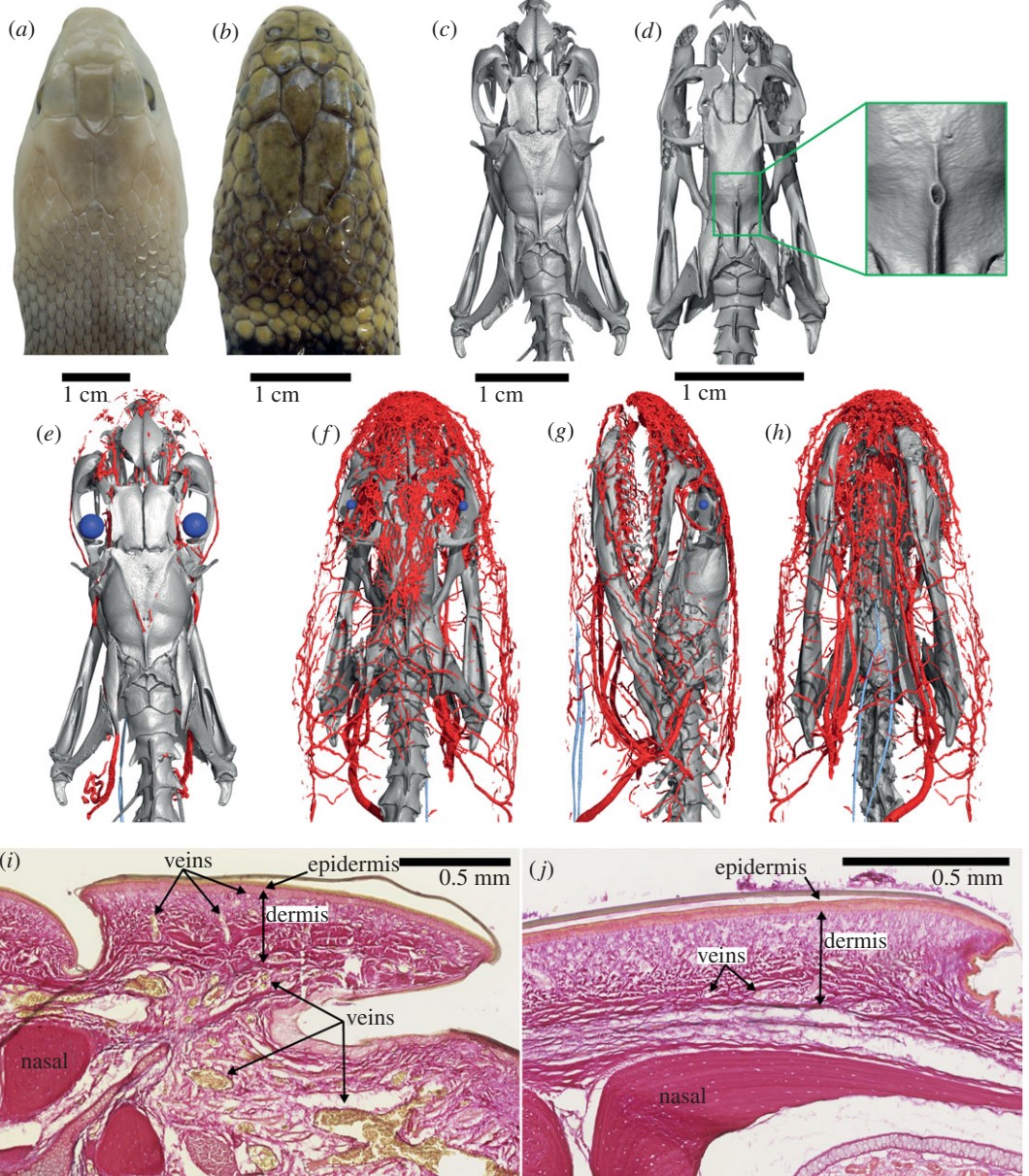

**Figure 1.** Heads of *Oxyuranus scutellatus* (*a,c,e*) and *Hydrophis cyanocinctus* (*b,d,f–h*), and histology sections (EVG) through the snout of the latter (*i*) and the former (*j*). Heads are shown in dorsal view, except in (*g*), left lateral and (*h*), ventral view. In (*e–h*), large blood vessels are coloured red and eye lenses blue. In the histology sections, erythrocytes (red blood cells) are stained yellow; note large blood vessels at the base of the dermis in *H. cyanocinctus*, and their smaller branches reaching just below the epidermis as capillaries. (*e*) is at the same scale as (*c*), and (*f–h*) are at the same scale as (*d*).

supplementary material, figure S2 in data S2). Histology sections revealed that the MCVN in *H. cyanocinctus* predominantly consists of veins and small sinuses. The largest blood vessels are located at the base of the dermis and branch dorsally into smaller blood vessels (capillaries) that are located immediately under the epidermis (electronic supplementary material, figure S3 in data S2). Inspection of the three-dimensional rendering of the segmented blood vessels (figure 1*f–h*) shows that the MCVN converges posteriorly towards a large blood vessel (paired in some specimens; figure 2*c*) that penetrates the skull via the unusually large parietal foramen observed in this species (figure 1*d*).

Two small parietal foramina can be observed in some elapids (e.g. *Oxyuranus*, figure 1*c*), and the large foramen observed in *H. cyanocinctus* clearly results from enlargement and merging of these foramina (some specimens of *H. cyanocinctus* still show a pair of adjacent foramina, figure 2). Histology sections stained for elastin (EVG) show that the blood vessel entering the braincase through the large foramen is a vein (thin tunica media) (figure 2*c* and electronic supplementary material, figure S4 in data S2)

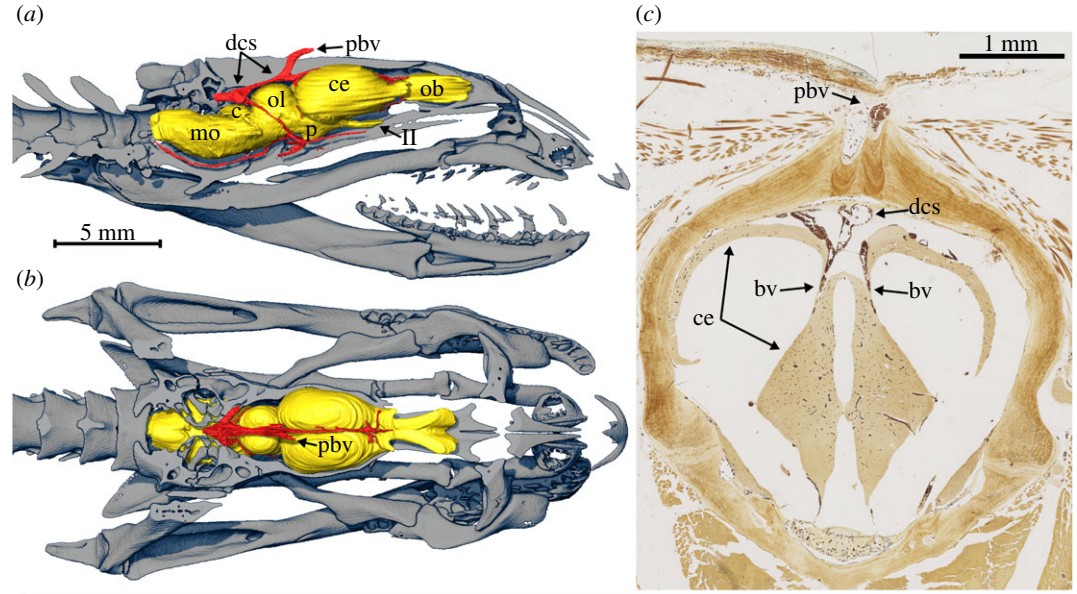

II, optic stalk; bv, blood vessel (vein); c, cerebellum; ce, cerebrum; dcs, dorsal cerebral sinus; mo, medulla oblongata; p, pituitary; pbv, parietal blood vessel; ob, olfactory bulb; ol, optic lobe.

**Figure 2.** Endocranial path of the cephalic vascular network (red) in *H. cyanocinctus* (brain coloured yellow). (*a*) Lateral view, right half of the skull removed. (*b*) Dorsal view, top half of the skull removed. (*c*) Stained histology cross section (Biel) at a level immediately anterior to the parietal foramen. Note pair of blood vessels (erythrocytes are stained dark brown) departing ventrally from the dorsal cerebral sinus and entering the brain. No nerve fibres exit the parietal foramen.

and that there is no nerve sharing its path (confirmed with Biel's stain, figure 2*c*). Endocranially, the vein in question leads to the dorsal cerebral sinus, and from there a symmetrical pair of large blood vessels departs ventrally and goes into the brain (figure 2*c*) (because the blood vessels are veins, we must assume that direction of flow is ventrally and caudally, i.e. towards the heart).

We formulated four alternative hypotheses for the function of the modified cephalic vascular network in *H. cyanocinctus* and discarded all but the fourth.

Hypothesis 1: the MCVN supplies cephalic mechanosensory organs called sensilla [13]. This hypothesis can be rejected because the distribution of sensilla on the head of this snake species does not match that of the blood vessels: the density of sensilla on the rostral scale does not appear to be higher than that on the labial scales (which are not very densely vascularized), and there are almost no sensilla on the frontal and parietal scales (which are densely vascularized). Moreover, a similarly dense network of nerve fibres would be expected to parallel the blood vessels, but this was not observed (figure 2*c*).

Hypothesis 2: the MCVN supplies blood to the erectile tissue that seals the nostrils underwater (haemostatic nostrils). This hypothesis can readily be rejected after considering that while all true sea snakes have haemostatic nostrils, only some possess an MCVN (electronic supplementary material, figure S1 in data S2). Importantly, a large parietal foramen was only observed in some sea snakes in the genus *Hydrophis*.

Hypothesis 3: the MCVN supplies a modified premaxillary gland (anterior portion of the supralabial gland) that has acquired the osmoregulatory function of a salt gland. Sea snakes have sublingually positioned salt glands but a modified premaxillary gland with salt excreting function has been reported in the independently aquatic homalopsid snake *Cerberus rhynchops* [14]. Salt glands are in fact known to be well vascularized [15]; however, they are predominantly serous (AB/PAS negative) [16], while the premaxillary gland in *H. cyanocinctus* is AB/PAS positive (electronic supplementary material, figure S5 in data S2). Moreover, the definite sublingual salt gland in *H. cyanocinctus* has a different structure (compact compound tubular) compared to its premaxillary gland, which is tubuloacinar (electronic supplementary material, figure S6 in data S2). A compound tubular structure appears to be a prerequisite of all vertebrate salt glands [17]. Last but not least, the sublingual salt gland in *H. cyanocinctus*, albeit well supplied with blood vessels, is not associated with a similarly

dense and complex network of large blood vessels such as that observed around the snout (premaxillary gland included) and forehead of the snake. Thus, also this hypothesis can be rejected.

Hypothesis 4: The MCVN provides an additional oxygen supply to the brain during submersion. Sea snakes obtain significant amounts of oxygen through their skin and a modified epidermal microstructure, i.e. mesos layer absent or very thin (electronic supplementary material, figure S2 in data S2), may further facilitate gas exchange through the skin via a reduced permeability barrier [12].

In shallow laboratory aquaria, with access to air, sea snakes use cutaneous oxygen uptake for up to 33% of the total [4,5], but it is likely that some marine hydrophiines that rest on the bottom for long periods can obtain 100% of their resting requirements [6]. Cutaneous oxygen uptake is only possible if the blood going to the skin is initially low in oxygen. Because the skin of all snakes is supplied with systemic arterial blood that has been through the lungs, it is typically high in oxygen. However, sea snakes such as *Hydrophis* have a unique circulation that largely bypasses the lungs and keeps the oxygen partial pressure ($PO_2$) low in the arterial blood, so that oxygen can diffuse through the skin from the seawater. Arterial $PO_2$ can consequently be extremely low in species of *Hydrophis*. For example, during voluntary dives, aortic $PO_2$ can decrease to 1.6 kPa in *Hydrophis (Acalyptophis) peronii* and 1.2 kPa in *Hydrophis curtus* (*Lapemis hardwickii*), values that are about 10% of those prevailing in the lungs, approximately 13 kPa [18], and much less than in seawater (approx. 20 kPa). The oxygen saturation of the haemoglobin remains near 60% and $PO_2$ below about 5 kPa, in *Hydrophis belcheri* and *Hydrophis ornatus*, despite applied pressures to simulate diving to 41 m, when lung $PO_2$ would rise toward 65 kPa [19].

Such low $PO_2$ in the arterial blood is potentially a problem for brain function, and *H. cyanocinctus* may have solved it by the unusual pattern of brain perfusion that derives high $PO_2$ blood from the MCVN rather than low $PO_2$ blood from the carotid arteries. We tested this hypothesis by measuring: (a) the total surface area of the skin covering the snout and forehead of the snake (199 mm$^2$); (b) the average thickness of the epidermis in this patch of skin (27 μm, average from 74 measurements taken on two histological cross sections) and (c) the volume of the brain (59 mm$^3$). When these numbers were used in Fick's diffusion equation with a diffusion coefficient for animal tissue [20,21] and a moderate difference in $PO_2$ across the skin (10 kPa), the result was a rate of diffusion of 0.001 ml min$^{-1}$ (electronic supplementary material, data S3). Assuming that the oxygen consumption of a whole resting sea snake is 0.038 ml min$^{-1}$ g$^{-1}$ [6], and the 59 mm$^3$ brain consumes oxygen at the same rate, then the MCVN would supply 25 times the brain's requirement. However, in mammals, brain metabolism is about 10 times the rate of the rest of the body [22]. If this were true for sea snakes, the MCVN would still supply three times the requirement of well-oxygenated blood. Such oversupply may be useful if the $PO_2$ difference across the skin drops.

## 4. Conclusion

In the light of the data presented above, the supply of additional oxygen to the brain during submersion provides the most plausible function for this unusual network of large blood vessels connected to the brain. If so, the MCNV would represent a novel adaptation. While only confirmed for *H. cyanocinctus*, it is likely to be present in other sea snakes that have distinctly enlarged parietal foramina, such as the closely related *H. spiralis* and *H. coggeri* but possibly also in the more distantly related *H. zweifeli* ([9]; A.P. personal observation, 2018). Assuming a direct correlation between a large parietal foramen and an MCVN, the disjunct occurrence of the foramen across the phylogeny of *Hydrophis* [23] would suggest that an MCVN may have been independently derived multiple times within the genus from the plesiomorphic elapid vascular system. However, despite the functional advantage provided by the MCVN, there is not yet any evident correlation between the presence of large parietal foramina (and likely associated MCVN) and particular ecologies of the snake species that possess them. The possibility that an MCVN may allow sea snakes to stay submerged for longer periods of time remains to be tested.

Ethics. This study did not involve human subjects or *in vivo* experiments. All specimens were opportunistically sourced and were euthanized in accordance with Animal Ethics Committee protocols from the University of Adelaide (S-2015-119).

Data accessibility. The electronic supplementary material, data S1–S3 contains details of the methods and results. The diceCT data from *H. cyanocinctus* is available from the Dryad Digital Repository: https://doi.org/10.5061/dryad. fq55cd7 [24].

Authors' contributions. A.P. conceived and designed the study, collected the data and wrote the first draft of the manuscript. R.S.S. conducted physiological assessment of the MCVN. C.V.N. acquired and fixed the specimens. All authors contributed to revising the manuscript and approved its final version.

Competing interests. The authors have no competing interests.

Funding. Financial support came from the Australian Research Council to M.L. and A.P. (grant no. DP 160103005) and KS (grant no. FT130101965) and Flinders University [capacity-building grant to A.P. and M.L.].

Acknowledgements. We thank Adelaide Microscopy and Microscopy Australia for access to the microCT scanning equipment at the University of Adelaide (Adelaide, Australia), and Ruth Williams for the assistance provided; Kathryn Batra, Agatha Labridinis and Emily Schneider from Histology Services at the Faculty of Health and Medical Sciences, University of Adelaide; Luke Allen, from Venom Supplies Ltd (Tanunda, South Australia), for providing some of the specimens used in this study; Marc E.H. Jones, Amy Watson and Jenna Crowe-Riddell for discussion; and last but not least, Harvey B. Lillywhite and an anonymous reviewer for helpful comments and suggestions.

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
