## [Reviewer comments · Royal Society Open Science]

Review History

RSOS-191099.R0 (Original submission)

Review form: Reviewer 1

Is the manuscript scientifically sound in its present form?

Yes

Are the interpretations and conclusions justified by the results?

Yes

Is the language acceptable?

Yes

Do you have any ethical concerns with this paper?

No

Have you any concerns about statistical analyses in this paper?

No

Recommendation?

Accept with minor revision (please list in comments)

Comments to the Author(s)

This is an exciting paper as it describes a novel, and entirely unexpected adaptation in sea snakes. The paper is well written, the methods valid, the illustrations expertly prepared and presented, and valid conclusions reached from the data. I recommend publication.

However, in the abstract, where the authors presumably are trying to minimise the number of words to fit within the prescribed limit, there are several constructions that should be revised.

They are:

Line 38 is teleological: I suggest changing "arterial blood to facilitate" to "arterial blood that facilitates"

Line 39: "oxygen" in a noun, not an adjective. I suggest changing "limiting oxygen availability to the brain" to "limiting availability of oxygen to the brain"

Line 41: "sea snake" is a noun, not an adjective. The possessive should be used. I suggest changing "sea snake cutaneous" to "sea snakes' cutaneous"

The body of the paper is relatively free of such constructions.

Review form: Reviewer 2 (Harvey B. Lillywhite)**Is the manuscript scientifically sound in its present form?**

Yes

Are the interpretations and conclusions justified by the results?

Yes

Is the language acceptable?

Yes

Do you have any ethical concerns with this paper?

No

Have you any concerns about statistical analyses in this paper?

No

Recommendation?

Accept with minor revision (please list in comments)

Comments to the Author(s)

This is an interesting and very well-written manuscript that reports a novel feature of vasculature in a sea snake. The approach of this investigation is conceptually sound, and the conclusions of the authors are well supported by the findings of the study. In my opinion, this manuscript represents an important, meritorious paper, and the authors have done a very nice job of reporting their findings. I have no suggestions for major changes, but I do have a number of queries and one editorial emendation that I feel might improve an already excellent paper. (Revisions related to the items below might not be feasible and are optional.)

1. Inasmuch as the vascular structure and its proposed function are not yet demonstrated to be an “adaptation for underwater respiration,” I think the title would be more informative if the words “novel adaptation for underwater respiration in” are changed to something like “novel vascular plexus in the head of”.
2. It is not clear from the descriptions and figures how far below the skin are the blood vessels of focus. Evidently these are deep in the dermis. Presumably there is a capillary network close to the epidermis, but the evidence for this is not discussed, nor are the “smaller branches” mentioned in the caption to Fig. 1 distinguishable to my eye (at least, near the epidermis). If the vascular network of larger vessels (MCVN) is indeed respiratory in function, one would expect connecting capillaries that are very close to the epidermis. I suggest the authors might address this issue.
3. Is there any further information concerning the ecology or behavior of this species that might be relevant to the discussion? This species is euryhaline, and there might be features of habitat where this snake occurs or sojourns that possibly relate to environmental oxygen levels, or reclusive behaviors of snakes that might expose them to variable oxygen levels. If there is any information about this, it would be significant to mention.
4. Is there external indication of the location of the foramen that is visible on the scales of the top of the head? I believe this might be so based on photos I have seen, but I am not sure.
5. Is there indication of anything special about the permeability barrier (mesos layer of epidermis; or arrangement of lipids) in the cephalic scales of this snake? This might be expected if the vascular system is specialized for gaseous exchange or uptake. Otherwise I would not expect the thickened scales of the head to be especially permeable.
6. Both items # 2 and 5 above are relevant to the use of Fick’s diffusion equation mentioned on page 8 of the manuscript (both diffusion distance and coefficient).

I do not wish to remain anonymous. – Harvey B. Lillywhite

Decision letter (RSOS-191099.R0)

22-Jul-2019

Dear Dr Palci

On behalf of the Editors, I am pleased to inform you that your Manuscript RSOS-191099 entitled "Novel adaptation for underwater respiration in a sea snake (Elapidae, Hydrophiinae) revealed by high-resolution computed tomography and histology." has been accepted for publication in Royal Society Open Science subject to minor revision in accordance with the referee suggestions. Please find the referees' comments at the end of this email.

The reviewers and handling editors have recommended publication, but also suggest some minor revisions to your manuscript. Therefore, I invite you to respond to the comments and revise your manuscript.

- Ethics statement

- Data accessibility

<http://datadryad.org/submit?journalID=RSOS&manu=RSOS-191099>

- Competing interests

- Authors' contributions

- Acknowledgements

- Funding statement

Please ensure you have prepared your revision in accordance with the guidance at <https://royalsociety.org/journals/authors/author-guidelines/> -- please note that we cannot publish your manuscript without the end statements. We have included a screenshot example of

the end statements for reference. If you feel that a given heading is not relevant to your paper, please nevertheless include the heading and explicitly state that it is not relevant to your work.

Because the schedule for publication is very tight, it is a condition of publication that you submit the revised version of your manuscript before 31-Jul-2019. Please note that the revision deadline will expire at 00.00am on this date. If you do not think you will be able to meet this date please let me know immediately.

Please note that Royal Society Open Science charge article processing charges for all new submissions that are accepted for publication. Charges will also apply to papers transferred to Royal Society Open Science from other Royal Society Publishing journals, as well as papers

submitted as part of our collaboration with the Royal Society of Chemistry (<http://rsos.royalsocietypublishing.org/chemistry>).

on behalf of Dr Jake Socha (Associate Editor) and Kevin Padian (Subject Editor)
openscience@royalsociety.org

Associate Editor Comments to Author (Dr Jake Socha):

Associate Editor: 1

Comments to the Author:

Congratulations on an excellent piece of scholarship. Both reviewers had glowing things to say about the manuscript, and we all agree that it is an exciting contribution to the literature. There are a few small suggestions from both reviewers. I would highly advise you to take those under consideration and revise the manuscript accordingly in the next submission, before final publication.

Reviewer comments to Author:

Reviewer: 1

Comments to the Author(s)

This is an exciting paper as it describes a novel, and entirely unexpected adaptation in sea snakes. The paper is well written, the methods valid, the illustrations expertly prepared and presented, and valid conclusions reached from the data. I recommend publication.

However, in the abstract, where the authors presumably are trying to minimise the number of words to fit within the prescribed limit, there are several constructions that should be revised. They are:

Line 38 is teleological: I suggest changing "arterial blood to facilitate" to "arterial blood that facilitates"

Line 39: "oxygen" in a noun, not an adjective. I suggest changing "limiting oxygen availability to the brain" to "limiting availability of oxygen to the brain"

Line 41: "sea snake" is a noun, not an adjective. The possessive should be used. I suggest changing "sea snake cutaneous" to "sea snakes' cutaneous"

The body of the paper is relatively free of such constructions.

Reviewer: 2

Comments to the Author(s)

This is an interesting and very well-written manuscript that reports a novel feature of vasculature in a sea snake. The approach of this investigation is conceptually sound, and the conclusions of the authors are well supported by the findings of the study. In my opinion, this manuscript represents an important, meritorious paper, and the authors have done a very nice job of reporting their findings. I have no suggestions for major changes, but I do have a number of queries and one editorial emendation that I feel might improve an already excellent paper. (Revisions related to the items below might not be feasible and are optional.)

1. Inasmuch as the vascular structure and its proposed function are not yet demonstrated to be an "adaptation for underwater respiration," I think the title would be more informative if the words "novel adaptation for underwater respiration in" are changed to something like "novel vascular plexus in the head of".
2. It is not clear from the descriptions and figures how far below the skin are the blood vessels of focus. Evidently these are deep in the dermis. Presumably there is a capillary network close to the epidermis, but the evidence for this is not discussed, nor are the "smaller branches" mentioned in the caption to Fig. 1 distinguishable to my eye (at least, near the epidermis). If the vascular network of larger vessels (MCVN) is indeed respiratory in function, one would expect connecting capillaries that are very close to the epidermis. I suggest the authors might address this issue.
3. Is there any further information concerning the ecology or behavior of this species that might be relevant to the discussion? This species is euryhaline, and there might be features of habitat where this snake occurs or sojourns that possibly relate to environmental oxygen levels, or reclusive behaviors of snakes that might expose them to variable oxygen levels. If there is any information about this, it would be significant to mention.
4. Is there external indication of the location of the foramen that is visible on the scales of the top of the head? I believe this might be so based on photos I have seen, but I am not sure.
5. Is there indication of anything special about the permeability barrier (mesos layer of epidermis; or arrangement of lipids) in the cephalic scales of this snake? This might be expected if the vascular system is specialized for gaseous exchange or uptake. Otherwise I would not expect the thickened scales of the head to be especially permeable.
6. Both items # 2 and 5 above are relevant to the use of Fick's diffusion equation mentioned on page 8 of the manuscript (both diffusion distance and coefficient).

I do not wish to remain anonymous. - Harvey B. Lillywhite

Author's Response to Decision Letter for (RSOS-191099.R0)

See Appendix A.

Decision letter (RSOS-191099.R1)

01-Aug-2019

Dear Dr Palci,

I am pleased to inform you that your manuscript entitled "Novel vascular plexus in the head of a sea snake (Elapidae, Hydrophiinae) revealed by high-resolution computed tomography and histology." is now accepted for publication in Royal Society Open Science.

on behalf of Dr Jake Socha (Associate Editor) and Kevin Padian (Subject Editor)
openscience@royalsociety.org

Appendix A

Reviewer comments to Author:

Reviewer: 1

Comments to the Author(s)

This is an exciting paper as it describes a novel, and entirely unexpected adaptation in sea snakes. The paper is well written, the methods valid, the illustrations expertly prepared and presented, and valid conclusions reached from the data. I recommend publication.

However, in the abstract, where the authors presumably are trying to minimise the number of words to fit within the prescribed limit, there are several constructions that should be revised. They are:

Line 38 is teleological: I suggest changing "arterial blood to facilitate" to "arterial blood that facilitates"

Line 39: "oxygen" in a noun, not an adjective. I suggest changing "limiting oxygen availability to the brain" to "limiting availability of oxygen to the brain"

Line 41: "sea snake" is a noun, not an adjective. The possessive should be used. I suggest changing "sea snake cutaneous" to "sea snakes' cutaneous"

The body of the paper is relatively free of such constructions.

We agree with all of Reviewer 1's comments and modified the abstract accordingly (edits highlighted in red font).

Reviewer: 2

Comments to the Author(s)

This is an interesting and very well-written manuscript that reports a novel feature of vasculature in a sea snake. The approach of this investigation is conceptually sound, and the conclusions of the authors are well supported by the findings of the study. In my opinion, this manuscript represents an important, meritorious paper, and the authors have done a very nice job of reporting their findings. I have no suggestions for major changes, but I do have a number of queries and one editorial emendation that I feel might improve an already excellent paper. (Revisions related to the items below might not be feasible and are optional.)

1. Inasmuch as the vascular structure and its proposed function are not yet demonstrated to be an "adaptation for underwater respiration," I think the title would be more informative if the words "novel adaptation for underwater respiration in" are changed to something like "novel vascular plexus in the head of".

We agree with the suggestion and modified the title accordingly.

2. It is not clear from the descriptions and figures how far below the skin are the blood vessels of focus. Evidently these are deep in the dermis. Presumably there is a capillary network close to the epidermis, but the evidence for this is not discussed, nor are the

“smaller branches” mentioned in the caption to Fig. 1 distinguishable to my eye (at least, near the epidermis). If the vascular network of larger vessels (MCVN) is indeed respiratory in function, one would expect connecting capillaries that are very close to the epidermis. I suggest the authors might address this issue.

We have added a sentence to the Results sections to better describe the presence and location of the blood vessels (lines 120-123). We have also added an additional supplementary figure, now Fig S3) to illustrate the capillaries located immediately under the epidermis.

3. Is there any further information concerning the ecology or behavior of this species that might be relevant to the discussion? This species is euryhaline, and there might be features of habitat where this snake occurs or sojourns that possibly relate to environmental oxygen levels, or reclusive behaviors of snakes that might expose them to variable oxygen levels. If there is any information about this, it would be significant to mention.

Unfortunately, there is no obvious correlation between ecology or behaviour and presence of a MCVN, as we stated on lines (227-231): “However, despite the functional advantage provided by the MCVN, there is not yet any evident correlation between the presence of large parietal foramina (and likely associated MCVN) and particular ecologies of the snake species that possess them.”

Like lots of *Hydrophis*, *H. cyanocinctus* is found in varied habitats that only sometimes include estuaries, whereas some *Hydrophis* do seem to be restricted to estuarine and inshore habitats (e.g. *H. obscurus*, *H. schistosus*, *H. caeruleus*, *H. donaldi*).

The challenges of hypoxia and cycling hypoxia-reoxygenation are probably common to all sea snakes. While *H. cyanocinctus* shares similar habitat types with other sea snakes, it might perform more strenuous swimming or predatory behaviours in these habitats. They can subdue large aggressive eels in deep water. But again so do several other *Hydrophis*.

4. Is there external indication of the location of the foramen that is visible on the scales of the top of the head? I believe this might be so based on photos I have seen, but I am not sure.

As shown in Fig. 1b, there is no indication of the presence of a foramen through the scales of the top of the head. We have now added a sentence to our Introduction (lines 63-65) to make this clear.

5. Is there indication of anything special about the permeability barrier (mesos layer of epidermis; or arrangement of lipids) in the cephalic scales of this snake? This might be expected if the vascular system is specialized for gaseous exchange or uptake. Otherwise I would not expect the thickened scales of the head to be especially permeable.

The histology sections suggest that the mesos layer in *H. cyanocinctus* is likely very thin or absent. A condition reminiscent of what has been recently documented in the congeneric

sea snake *H. platurus* (Lillywhite and Menon, 2019). We have now added two sentences (lines 116-119 and lines 176-178) to address this point, added the reference in question (ref. 12), and also added a new supplementary figure (Fig. S2) to illustrate the difference between the epidermis of *Oxyuranus* (a terrestrial snake with a well-developed mesos layer) and *H. cyanocinctus* (where the mesos layer is absent or very thin).

6. Both items # 2 and 5 above are relevant to the use of Fick's diffusion equation mentioned on page 8 of the manuscript (both diffusion distance and coefficient).

We have addressed both points (see above).

I do not wish to remain anonymous. – Harvey B. Lillywhite